# Platelet Microparticles Decrease Daunorubicin-Induced DNA Damage and Modulate Intrinsic Apoptosis in THP-1 Cells

**DOI:** 10.3390/ijms22147264

**Published:** 2021-07-06

**Authors:** Daniel Cacic, Oddmund Nordgård, Peter Meyer, Tor Hervig

**Affiliations:** 1Department of Hematology and Oncology, Stavanger University Hospital, 4068 Stavanger, Norway; oddmund.nordgard@sus.no (O.N.); peter.albert.meyer@sus.no (P.M.); 2Department of Clinical Science, University of Bergen, 5021 Bergen, Norway; tor.audun.hervig@helse-fonna.no; 3Laboratory of Immunology and Transfusion Medicine, Haugesund Hospital, 5528 Haugesund, Norway

**Keywords:** acute myelogenous leukemia, platelets, microparticles, apoptosis

## Abstract

Platelets can modulate cancer through budding of platelet microparticles (PMPs) that can transfer a plethora of bioactive molecules to cancer cells upon internalization. In acute myelogenous leukemia (AML) this can induce chemoresistance, partially through a decrease in cell activity. Here we investigated if the internalization of PMPs protected the monocytic AML cell line, THP-1, from apoptosis by decreasing the initial cellular damage inflicted by treatment with daunorubicin, or via direct modulation of the apoptotic response. We examined whether PMPs could protect against apoptosis after treatment with a selection of inducers, primarily associated with either the intrinsic or the extrinsic apoptotic pathway, and protection was restricted to the agents targeting intrinsic apoptosis. Furthermore, levels of daunorubicin-induced DNA damage, assessed by measuring gH2AX, were reduced in both 2N and 4N cells after PMP co-incubation. Measuring different BCL2-family proteins before and after treatment with daunorubicin revealed that PMPs downregulated the pro-apoptotic PUMA protein. Thus, our findings indicated that PMPs may protect AML cells against apoptosis by reducing DNA damage both dependent and independent of cell cycle phase, and via direct modulation of the intrinsic apoptotic pathway by downregulating PUMA. These findings further support the clinical relevance of platelets and PMPs in AML.

## 1. Introduction

Platelets were originally discovered in the late 19th century as a key player in hemostasis [1]. It is now clear, however, that they serve a broader role in both health and disease [2,3,4,5,6,7]. Platelets contain many different biologically active molecules, which include proteins [8,9], regulatory microRNAs [10,11], and long RNA sequences, such as ribosomal RNAs and protein-coding transcripts inherited from parental megakaryocytes [12,13]. The long RNA sequences are prone to time-dependent decay [12,14], and correlation with the proteome is weak [13], suggesting only a limited protein synthesis capacity, which may be confined to reticulated platelets [12].

Bioactive substances can be secreted from platelets as paracrine or endocrine factors that are able to modify various cancers [15,16,17]. These bioactive molecules can also be transferred via platelet microparticles (PMPs), which in turn have been shown to be internalized by many different cancer cells, altering crucial functions of the cells, namely invasiveness, proliferation, and viability [18,19,20]. The pro-tumoral properties of platelets are further supported by retrospective and observational data showing an association between platelet inhibition and decreased risk for development of cancer, and increased cancer-specific survival [21,22,23,24]. However, the mechanism underlying this potential effect remains unknown, and the data from the few prospective studies that have been done are less convincing [25,26,27].

Acute myelogenous leukemia (AML) is a bone marrow disease affecting hematopoietic stem and progenitor cells [28,29]. The genomic landscape of AML has been thoroughly analyzed and the first study performing whole-genome sequencing in AML was already published in 2008 [30]. AML usually has a lower frequency of somatic mutations than most other cancers [31,32], with a median of 13 different coding mutations per case [33,34]. Despite the low mutational burden, there are large variations in transcriptomic and proteomic signatures in AML cells, compared with healthy hematopoietic progenitors, and even between different subclones [35,36,37]. Despite increased knowledge in the genomics of AML, treatment strategies have essentially remained unchanged for several decades, with a few exceptions [38]. Curative treatment, which is restricted to younger patients, consists of intensive chemotherapy with consolidating hematopoietic stem cell treatment for high-risk cases. Despite this, median survival is just 11 months when including all age groups [39], underscoring the need for a more profound understanding of progression of the disease and development of treatment resistance, in addition to established genomic mechanisms.

Targeting apoptosis in cancer is a novel treatment strategy that is finally maturing into clinical use. Apoptosis can be divided into two separate pathways, which are interlinked with common feedback mechanisms [40]; the death receptor initiated extrinsic pathway (FAS/CD95, TNFR1, TRAIL-R1, TRAIL-R2, DR3, and DR6), and the intrinsic, or mitochondrial, pathway. Dysregulation of the latter has proven to be an important feature in cancer biology [41]. The regulatory and anti-apoptotic proteins in the BCL2-family are also known to be upregulated in hematological malignancies [42,43]. Hence, numerous drugs that target major apoptotic regulators, such as BCL2 or MCL1, are currently either under development, or have just been approved, to treat a variety of hematological malignancies, including AML [41].

The intrinsic apoptotic pathway is initiated by several factors, including DNA damage or cellular stress, which is accompanied by upregulation of the pro-apoptotic BH3-only proteins (including BAD, BID, NOXA, HRK, BMF, PUMA, BIM), which then activate the effector proteins BAK and BAX directly or through inhibition of anti-apoptotic regulator proteins [44,45]. Upon activation, the predominantly mitochondrial outer membrane (MOM)-bound BAK, and predominantly cytosolic BAX protein, oligomerize in the MOM, leading to cytochrome c leakage from the mitochondria [46,47]. Cytochrome c then forms an apoptosome with apoptotic protease activating factor-1 (APAF1), which recruits pro-caspase 9, both activating and regulating its function [48,49]. Caspase-9 activates caspase-3, where the intrinsic and extrinsic pathways converge. Caspase-3 has multiple substrates [50], including a caspase-dependent DNase, which leads to DNA degradation upon activation by caspase-3 [51].

Our group has previously shown that co-incubation of the monocytic AML cell line, THP-1, or primary AML samples, with platelet microparticles, protects against daunorubicin (DNR)-induced apoptosis and cell death, at least partially via a decrease in cell activity [52]. We also found that miR-125a and miR-125b levels were elevated in THP-1 cells after PMP co-incubation. These microRNAs have been associated with chemotherapy resistance [53,54]. However, whether the PMP-associated increase in resistance to DNR is caused solely by protection against DNR-induced cell damage, or a modulation of the intrinsic apoptotic pathway regulators, remains unknown. Thus, we sought to further examine the anti-apoptotic effects of PMPs in the monocytic AML cell line THP-1.

## 2. Results

### 2.1. PMPs Offered Protection from Apoptosis Induced by Multiple Agents

We have previously demonstrated that PMPs increase resistance to DNR-induced apoptosis and cell death [52]. To investigate whether co-incubation with PMPs provided a general anti-apoptotic effect, we compared apoptosis and cell death after treatment with several agents associated with inducing apoptosis, primarily through intrinsic (alantolactone, staurosporine, MG 132), or extrinsic (piceatannol, TRAIL) apoptosis. Co-incubation of PMPs with THP-1 cells decreased the relative frequency of dead and apoptotic cells induced by alantolactone, staurosporine, and MG 132, but not piceatannol (Figure 1). In our experiments 50 ng/mL TRAIL was not sufficient to induce apoptosis in THP-1 cells, but it slightly potentiated the apoptotic effect of piceatannol. Surprisingly, PMP co-incubation increased the relative frequency of dead and apoptotic cells in the case of the combination of piceatannol and TRAIL (*p* = 0.003). However, as this effect was marginal (mean difference of 1.89 percentage points; SD 0.43), it could be biologically insignificant. From these analyses, we suggest that PMPs may provide general protection from apoptosis, but seemingly only against agents that primarily activate the intrinsic apoptotic pathway.

### 2.2. PMPs Reduced Both Caspase-8 and Caspase-9 Activation Induced by DNR

The cytotoxic effect of DNR is commonly associated with an increase in DNA damage, i.e., an intrinsic stimulus. However, it is also suggested to activate the extrinsic apoptotic pathway [55]. To evaluate activation of intrinsic and extrinsic apoptosis after DNR-treatment, we measured levels of active caspase-8 and caspase-9 by flow cytometry, and gated the cells in “lo”, “mid”, and “hi” populations. In the case of caspase-8, it was not possible to accurately discriminate between the “mid” and “hi” populations, and consequently these populations were gated as one. Our analyses indicated that both caspases were highly activated after DNR-treatment, but this was partially inhibited by PMP co-incubation (Figure 2A, Appendix A). In addition, fluorescence of the respective caspases were decreased for all subpopulations in PMP co-incubated cells (Figure 2B, Appendix A). The relative decrease in frequency of caspase-8 or caspase-9 “mid/hi” cells associated with PMP co-incubation were equal (Figure 2C; *p* = 0.756). These findings indicated that activation of caspase-8 is important in DNR-induced apoptosis, and is most likely inhibited by PMPs via an upstream mechanism common with caspase-9 activation.

### 2.3. PMP Co-Incubation Downregulated Pro-Apoptotic PUMA Protein

To further investigate if PMPs could independently affect intrinsic apoptosis, we analyzed levels of BCL2-family proteins with and without PMP co-incubation, and both with and without DNR. Gating strategy is summarized in Figure 3. For the cell population only visible with DNR-treatment (P2), levels of BAK, BCL2, MCL1, and PUMA were relatively less increased with PMP co-incubation (Figure 4), when compared to non-DNR-treated THP-1 cells (P1), and the decrease seen in BMF levels was relatively less. We also identified the P1 population in DNR-treated cells, and antibody fluorescence intensity was more or less unaffected, except for BMF, which had a somewhat higher level than the P1 population in non-DNR-treated cells. The relative change in fluorescence intensity accompanying PMP co-incubation was as anticipated, and followed the expected trend of protection from DNR-induced cell damage with PMP co-incubation. For example, the fluorescence intensity of BAK increased with DNR in both groups, but the increase was less with PMP co-incubation than without. However, in the case of PUMA we found a reduced signal intensity with PMP co-incubation in all cell populations, independent of DNR. Thus, we suggest that PMP co-incubation may protect THP-1 cells against DNR-induced cell death, at least partially through downregulation of the pro-apoptotic PUMA protein.

### 2.4. Inhibitors of Caspase-9 and BAX Protected Against DNR-Induced Cell Death, but Less so with PMP Co-Incubation

As PMPs can decrease PUMA protein levels, DNR-induced apoptosis in cells co-incubated with PMPs may be less driven by the intrinsic apoptotic pathway. We investigated whether the protective effect of two inhibitors of intrinsic apoptosis, iMAC1 (BAX) and Q-LEHD-Oph (caspase-9), was affected by PMP co-incubation prior to DNR-treatment. We found a lower relative reduction in the relative frequency of dead and apoptotic cells, both for iMAC1 and Q-LEHD-Oph, with PMP co-incubation, which may indicate that caspase-9 activation was a weaker driver in apoptosis (Figure 5A). In addition, inhibiting the activity of caspase-9 or BAX with Q-LEHD-Oph and iMAC1 only yielded a reduction in levels of active caspase-9 in the “NO PMP” setting (Figure 5B). Thus, inhibitors of the intrinsic apoptotic pathway were less effective when THP-1 cells were co-incubated with PMPs, suggesting a direct modulation of this pathway.

### 2.5. PMP Co-Incubation Reduced DNA Damage After DNR-Treatment Independently of Cell Cycle Phase

To evaluate the anti-apoptotic effect of PMP co-incubation, we indirectly analyzed double-stranded DNA-breaks (DSBs) through measurement of phosphorylated histone H2AX, or gH2AX, after four h of DNR-treatment. As the process of apoptosis increases DSBs, we first investigated if apoptosis was induced within this time frame. We found that after four h apoptosis was still at the baseline level (Figure 6A). As expected, fluorescence of gH2AX was increased after DNR-treatment in 4N cells compared to 2N cells for both groups, and the relative frequency of 2N cells was increased with PMP co-incubation (Figure 6B,C). Additionally, the fluorescence of gH2AX was decreased, both for 4N cells, and more surprisingly, for 2N cells with co-incubation of PMPs, compared to the “NO PMP” setting (Figure 6B). These findings indicated that PMP co-incubation protected THP-1 cells against DNR-induced apoptosis by decreasing the amount of DNA damage produced by DNR-treatment, both dependently and independently of cell cycle inhibition.

## 3. Discussion

Platelets are now recognized as an important contributor in cancer biology through several mechanisms involving immune evasion, metastasis, and development of cancer microenvironments [15,56,57,58,59,60]. We have previously shown that platelet microparticles increase resistance to DNR in acute myelogenous leukemia cells as a result of decreasing cell activity [52]. Here we provide evidence that this effect is multifactorial. We showed that PMPs protected equally against caspase-8 and caspase-9 activation in DNR-induced apoptosis, and that PMPs decreased DNR-induced DNA damage, not just by inhibiting cell cycle progression. The PMPs also directly modulated intrinsic apoptosis via the downregulation of the pro-apoptotic PUMA protein.

The anti-apoptotic effect of PMPs was evident with alantolactone, staurosporine, and MG 132, all primarily associated with activation of the intrinsic apoptotic pathway in THP-1 cells [61,62,63]. On the other hand, PMPs did not protect THP-1 cells against piceatannol or a combination of TRAIL + piceatannol, which are known to activate death receptor 5 and the extrinsic apoptotic pathway [64]. This indicates that PMPs may have broader anti-apoptotic properties, albeit restricted to intrinsic apoptosis. However, it yields no insight into the distinct mechanisms, which could be a common upstream effect on the intrinsic apoptotic pathway, e.g., cell cycle inhibition decreasing induced cellular stress or DNA damage. Interestingly, MG 132 has been shown to induce apoptosis in THP-1 cells arrested in either G1 or G2/M phases, but not when macrophage differentiation is induced [63]. Previously we have shown that PMPs inhibit cell cycle progression, and stimulate differentiation towards macrophages [52]. Thus, the latter could represent an anti-apoptotic mechanism independent of cell cycle inhibition by PMPs.

Apoptosis induction by DNR is generally believed to be a result of inhibition of topoisomerase II (Top2) enzyme activity, leading to a rise in DNA-Top2 cleavage complexes [65]. The dependence of intact p53 protein for apoptosis induction by doxorubicin, a related Top2 poison, suggests a strong reliance on the activation of the intrinsic apoptotic pathway for this class of chemotherapeutics [66]. However, there is evidence that DNR-treatment upregulates death receptors and activate caspase-8 in multiple leukemic cell lines, thereby inducing extrinsic apoptosis [55]. In our experiments, caspase-8 and caspase-9 activation was equally inhibited by PMPs, suggesting that PMPs interfere with a common activation mechanism. However, this does not completely rule out a skew in upstream initiation of these pathways, as the levels of active caspase-8 and caspase-9 are also regulated by the downstream caspase-3 and caspase-7 as important feedback mechanisms [40].

We showed that not only was the relative frequency of caspase-9 positive cells lower with PMP co-incubation, but the potency of caspase-9 and BAX inhibitors was also reduced. Both these findings suggest a weaker drive from the intrinsic apoptotic pathway in PMP co-incubated cells, but also correlated with a reduction in the ultimate function of these molecules, which is the inhibition of caspase-9 activation. iMAC1 inhibits conformational activation of BAX, and maybe BAK, without competing with BH3 only proteins [67,68]. The anti-apoptotic effect of iMAC1 is also known to decrease with higher levels of BAX [69], but this should increase chemosensitivity [70], which is the opposite of the effects associated with PMPs. We suggest a common mechanism to explain the reduction in potency of both inhibitors. LEHD (leu-glu-his-asp)-sequence based peptides block the catalytic activity of caspase-9 [71]. iMAC1 will also lead to a decrease in caspase-9 activity by inhibiting mitochondrial outer membrane permeabilization [72]. Thus, both inhibitors ultimately lead to a decrease in the activation of caspase-3, which is not only essential for apoptosis induction, but also for caspase-9 activation in a feedback loop [40]. Thus, if the intrinsic apoptotic pathway is inhibited by PMP co-incubation, the relative contribution of this pathway to caspase-3 activation is reduced compared to the extrinsic pathway, which is also activated by DNR. This should lead to a relative reduction in efficiency of apoptosis inhibition through the intrinsic apoptotic pathway, as extrinsic apoptosis is presumably unaffected by both PMPs and the inhibitors. However, one important caveat for this conclusion is the selectivity of the caspase-inhibitor, which, at least in the older generation inhibitors, is proven to be poor [71]. There are some indications that the second generation inhibitor Q-LEHD-Oph also inhibits caspase-8, but this has not been analyzed in a cell-free system and it was less extensive then the caspase-9 inhibition [73]. Furthermore, our conclusion is supported by results involving two independent inhibitors of separate stages in the intrinsic apoptotic pathway.

The inhibitory effect of PMPs on cell cycle progression is a possible mechanism for increased DNR-resistance, since Top2 poisons are believed to be most effective in proliferative cells [65]. We have previously provided evidence for this, showing that serum starvation of THP-1 cells significantly reduces DNR-induced apoptosis [52]. By measuring gH2AX we showed that PMP co-incubation decreased the level of DNA damage after DNR-treatment. Phosphorylation of histone H2AX is induced by double-stranded DNA-breaks as a DNA damage response [74]. Thus, the level of gH2AX is a widely used proxy for DSBs in biological research [75,76,77]. As expected, gH2AX levels increased more in dividing 4N cells (G2/M), compared with non-dividing 2N cells (G1) across both groups. As PMPs inhibit cell cycle progression, this would necessarily decrease the level of DNA damage. However, we identified a relative decrease in the signal intensity of gH2AX with PMP co-incubation for both cell phases, suggesting a de facto protective mechanism against the effects of DNR, independent of cell cycle inhibition. Somewhat surprisingly, gH2AX levels were also lower with PMP co-incubation in cells in the G1 cell phase. Previously we have found a decrease in mitochondrial membrane potential associated with PMP co-incubation [52], which may decrease the level of reactive oxygen species (ROS). Significant DNA damage in cells in G1 cell phase is also found in doxorubicin-treated U2OS osteosarcoma cells [78]. This probably has a different etiology compared to the mechanism in the G2/M cell phase and may be explained by an increase in ROS [79,80].

An important question regarding the anti-apoptotic effect of PMPs is if they can directly modulate the apoptotic response. We measured anti-apoptotic (BCL2 and MCL1) and pro-apoptotic (BAK, BMF, and PUMA) BCL2-familiy proteins, both in response to DNR and at baseline, in a “NO DNR” setting. The decreased levels of PUMA associated with PMP co-incubation probably represent a de facto downregulation, as it was present in all cell populations both with and without DNR. This could be due to increased levels of microRNAs, miR-125a and miR-125b, which are transferred by PMPs [52], and proven to downregulate the protein at the translational level, inducing chemoresistance [53,54]. Furthermore, the “readiness” for activation of intrinsic apoptosis in AML cells has clinical relevance as it is a predictor of outcome with conventional treatment [81]. The other proteins analyzed were also altered, but not in the viable, non-DNR-treated cells, and always in sync with an expected decrease in apoptosis and cell damage associated with PMP co-incubation. Thus, it cannot be stated that these proteins were directly downregulated as a result of PMP-internalization. These differences could be a result of an altered regulation of BCL2-family proteins caused by downregulation of other proteins, such as PUMA. However, they may also stem from a shift in ratio of apoptotic to dead cells, which we did not discriminate. Surprisingly, DNR increased the fluorescence intensity for both the anti-apoptotic proteins tested (BCL2 and MCL1); one would expect a decrease in the level of anti-apoptotic proteins when apoptosis is induced. However, this pattern has been observed for some anti-apoptotic proteins in select leukemic cell lines and is presumably transitory [82].

PUMA is regulated by several factors, including different transcription factors and proteins like forkhead box O (FOXO) and p53 family members [83]. However, these mechanisms may be shared with other pro-apoptotic BCL2-familiy proteins [84], and therefore do not coincide with our observations of isolated PUMA downregulation. PUMA is also post-translationally regulated by phosphorylation and proteosomal degradation, which is proven to be induced by interleukin-3 and HER2 [85,86], but none of these proteins are considered to be a part of the platelet granule or releasate proteome [8,9]. In addition, there are other microRNAs that are present in PMPs, like miR-221 and miR-222 [11], which also are known to downregulate PUMA [87]. However, this has not been investigated in THP-1 or other acute myelogenous leukemia cell lines.

The evidence provided here supplements our previously published work that PMPs have anti-apoptotic properties in acute myelogenous leukemia. This effect could stem partially from inhibition of cell cycle progression and cell activity, making the cells less susceptible to damage induced by chemotherapy. In addition, we showed that PMPs may modulate the intrinsic apoptotic pathway through downregulation of PUMA, as a mechanism independent of cell cycle inhibition. The mechanistic findings from this study were derived solely from one cell line and need to be confirmed in primary AML cells. Nonetheless, translational research with PMPs in AML is warranted, as the indications for platelet inhibition to decrease PMP production, and thus potentially increase chemosensitivity, are further supported.

## 4. Materials and Methods

### 4.1. Cell Line

The THP-1 cell line was purchased from the American Type Culture Collection (Manassas, VA, USA) and cultured in Iscove’s Modified Dulbecco’s Medium (IMDM; Thermo Fisher Scientific, Waltham, MA, USA) + 10% FBS (Sigma Aldrich, St. Louis, MO, USA). Culture medium was partially replaced approximately every second day to keep the total cell concentration in the range of 2–6 × 10^5^ per mL, and cells were only used in experiments once the exponential growth phase was reached. Cultures were kept for less than three months.

### 4.2. Platelet Concentrate

Platelet concentrates pooled from four donors were produced using the automated Tacsi system (Terumo BCT, Lakewood, CO, USA) and were provided by the Department of Immunology and Transfusion Medicine, Stavanger University Hospital (Stavanger, Norway). The platelet concentrations were 0.94–1.06 × 10^9^ per mL. Leukocytes were removed by filtration to a residual level of <1.00 × 10^6^. The storage medium for the platelets was approximately 35% plasma and 65% additive solution (PAS-III; Baxter, Lake Zurich, IL, USA). Written consent was obtained from all donors.

### 4.3. Platelet Releasate

Platelet releasate was produced by adding human thrombin (Sigma Aldrich) at a final concentration of 1 U/mL to the platelet concentrates in 50 mL tubes, and incubating for one hour in a 37 °C water bath, as described in [52]. The releasates were mixed by gentle shaking every 5 min. To separate the releasate from the clot, the tubes were centrifuged for 10 min at 900 g and the supernatant was transferred to new 50 mL tubes. The samples were stored at −80 °C. Fibrin clots that appeared after thawing were removed using a 10 mL serological pipette.

### 4.4. Platelet Microparticle Production

Platelet microparticles were isolated as previously described [52]. Briefly, platelet releasate was centrifuged at 15,000 g for 90 min at room temperature, and the supernatant carefully poured off. The PMPs were then resuspended in IMDM + 10% FBS and transferred to cell culture, thoroughly mixing with the cells by pipetting. The final concentration of PMPs was 1.5 × 10^7^ per mL culture medium in all experiments. The wells were mixed again by pipetting 2 h after the PMPs were added to the cell cultures.

### 4.5. Platelet Microparticle Quantitation

One mL of platelet releasate, washed with 9 mL of Dulbecco’s phosphate-buffered saline (Sigma Aldrich), was centrifuged as described above and the supernatant carefully poured off. The platelet microparticles were resuspended in 400 µL of 0.22 µm filtered Annexin V Binding Buffer (Miltenyi Biotec, Bergisch Gladbach, Germany), and 200 µL of the solution was transferred to a second tube. The solution was then stained with 20 µL of Annexin V FITC (Milteny Biotec), and 2 µL of anti-CD61 APC (clone Y2/51; Miltenyi Biotec), or 22 µL of 0.22 µm filtered Annexin V Binding Buffer for a negative control and incubated for 15 min at room temperature. Finally, 278 µL of 0.22 µm filtered Annexin V Binding Buffer and 50 µL CountBright beads (Thermo Fisher Scientific) were added before analysis. Microparticle gates were set using Megamix-PLUS FSC beads (bead size range: 0.3 to 0.9 µm; BioCytex, Marseille, France), according to our previous report [52]. At least 2500 bead events were collected. This, and all other flow cytometric analyses, were performed on a CytoFLEX flow cytometer (Beckman Coulter, Brea, CA, USA) using CytExpert ver. 2.4 acquisition and analysis software (Beckman Coulter). An example of gating strategy for PMP quantitation can be found in Appendix A (see Appendix A).

### 4.6. Apoptosis Assay

Approximately 5 × 10^5^ cells per mL THP-1 cells were cultured with or without PMPs for 24 h. The cells were then treated with an apoptosis inductor at a concentration and time interval as indicated in Appendix A. Cell viability was analyzed with the Annexin V FITC Kit (Miltenyi Biotec), strictly following the manufacturer’s instructions. Dead and apoptotic cells were analyzed using flow cytometry and gated out in a single gate using a dot plot of FITC-A versus PerCP Cy 5.5-A after doublet discrimination. At least 20,000 gated cells were collected. An example of gating strategy can found in Appendix A.

### 4.7. Apoptosis Inhibition

For select experiments, after the initial 23 h of incubation with or without PMPs, the THP-1 cells were pretreated with either 20 µM of the caspase-9 inhibitor, Q-LEHD-Oph (Abcam, Cambridge, UK), or 10 µM of the BAX inhibitor, iMAC1 (Sigma Aldrich), and incubated for one hour before adding DNR, as described in the previous section.

### 4.8. Caspase Activity

Caspase-8 and caspase-9 activity in THP-1 cells after 24 h of DNR-treatment was measured using the CaspGLOW Fluorescein Active Staining Kit for the respective caspases (Thermo Fisher Scientific). Approximately 5 × 10^5^ cells in 0.3 mL IMDM + 10% FBS were incubated with 1 µL of either FITC-IETD-FMK or FITC-LEHD-FMK for 30 min in a CO_2_ incubator before washing twice with the supplied wash medium, and analyzing with flow cytometry. Both untreated and treated, but not stained, THP-1 cells were used as controls to determine low, medium, and high caspase populations. At least 20,000 gated cells were collected.

### 4.9. gH2AX

Measurement of gH2AX by flow cytometry was performed according to Darzynkiewicz et al. [75]. After PMP co-incubation, cells were fixed with 1% methanol free formaldehyde for 15 min on ice, then fixed and permeabilized in 70% ethanol, and stored overnight. Fixed and permeabilized cells were stained with FITC conjugated anti-phospho-Histone H2A.X (Ser139) antibody (clone JBW301; 1 µg/100 µL; Sigma Aldrich), and propidium iodide solution (5 µg/mL; Thermo Fisher Scientific) containing DNase free RNASE A/T1 cocktail (25 U/1000 U per mL; Thermo Fisher Scientific).

### 4.10. BCL2-Family Proteins

THP-1 cells were cultured for 24 h with or without PMPs, and an additional 24 h with or without treatment with DNR, before analysis of intracellular proteins using the published protocol by Ludwig et al. [88]. Briefly, cells were fixed and permeabilized using the eBioscience Foxp3/Transcription Factor Staining Buffer Set (Thermo Fisher Scientific). Cells were then incubated with unconjugated antibodies and labeled with the proper conjugated secondary antibodies. A list of the antibodies used, dilutions, and incubation times, can be found in Appendix A. A “no primary antibody” sample was used to subtract the background signal. At least 25,000 gated cells were collected.

### 4.11. Statistical Analyses

Statistical analyses were performed using the IBM SPSS 26 software (IBM Corp, Armonk, NY, USA). All figures show mean values with 95% confidence intervals. A comparison of means was performed using tests for paired data, or one-sample tests, when appropriate. The data were checked for normality using PP plots, the Shapiro–Wilks test, and the Kolmogorov–Smirnov test. A *p* value < 0.05 was considered significant. “*n*” denotes technical replicates.

## Figures and Tables

**Figure 1 ijms-22-07264-f001:**
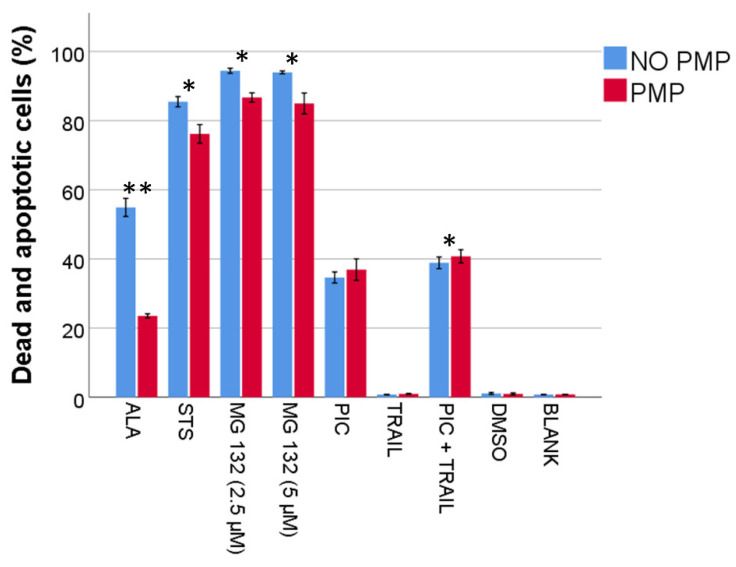
Apoptosis inhibition by platelet microparticles (PMPs). THP-1 cells with or without PMP co-incubation for 24 h were treated with an apoptosis-inducing molecule at a concentration and an incubation time as described in Appendix A (*n* = 4). Relative frequency of dead and apoptotic cells were analyzed by flow cytometry, and gated out in a single gate (annexin V vs. propidium iodide). Data were compared using the paired-sample *t*-test for data pairs. * *p* < 0.05, ** *p* < 0.001. ALA, alantolactone. STS, staurosporine. Pic, piceatannol. TRAIL, tumor necrosis factor-related apoptosis-inducing ligand.

**Figure 2 ijms-22-07264-f002:**
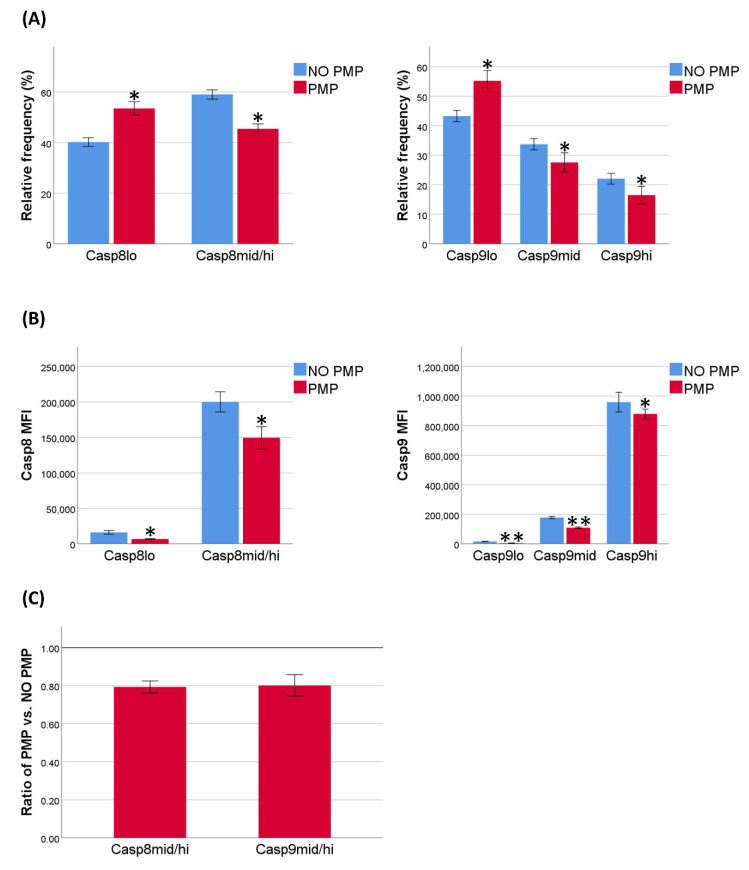
Caspase-8 and caspase-9 activation after daunorubicin (DNR)-treatment. (**A**) Active caspase-8 and caspase-9 were analyzed by flow cytometry after 24 h, with or without platelet microparticle (PMP) co-incubation, and an additional 24 h with DNR-treatment at 0.5 µM (*n* = 4). Events were gated as either “lo”, “mid”, or “hi”, according to fluorescence intensity. Data are presented as the relative frequencies of the populations. (**B**) Background corrected mean fluorescence intensity (MFI) of the respective caspases for different populations. (**C**) Ratio of the relative frequencies of the subpopulations “mid” and “hi” combined for respective caspases between THP-1 cells, with versus without PMP co-incubation. Data were compared using the paired-sample *t*-test for data pairs. * *p* < 0.05. ** *p* < 0.001. Casp8, caspase-8. Casp9, caspase-9.

**Figure 3 ijms-22-07264-f003:**
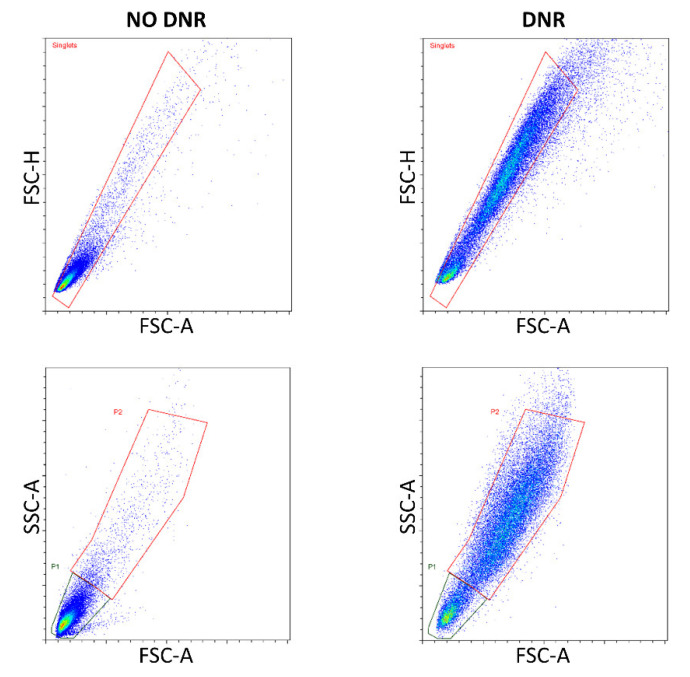
Gating strategy for intracellular flow cytometry of BCL2-family proteins. Doublets were first discriminated in FSC-A vs. FSC-H plots. P1 represents the population of daunorubicin (DNR)-treated cells gated in FSC-A vs. SSC-A plots that aligned well with non-DNR-treated cells. P2 represents a population generated by DNR-treatment and with increased light scatter.

**Figure 4 ijms-22-07264-f004:**
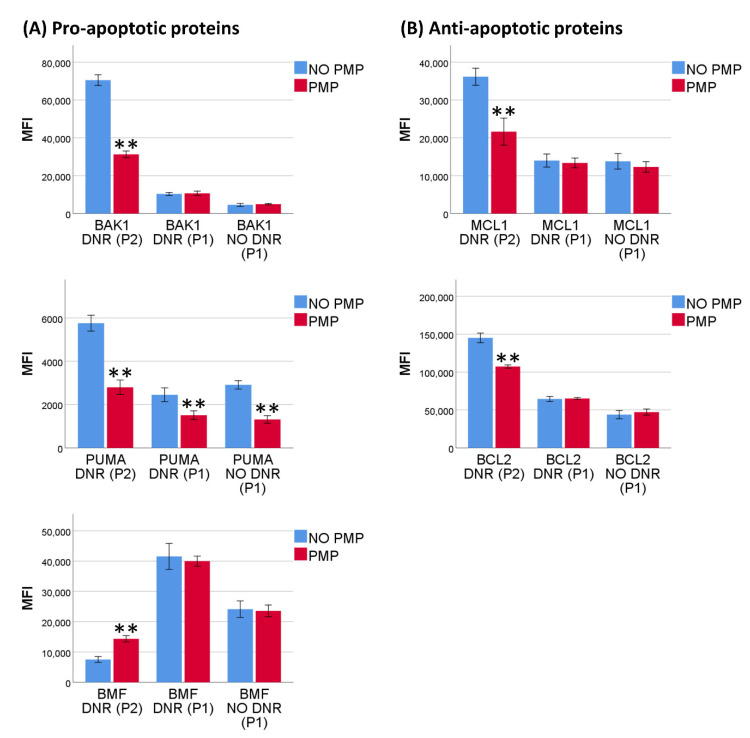
Levels of BCL2-family proteins before and after daunorubicin treatment. (**A**) THP-1 cells were co-incubated with or without platelet microparticles (PMPs) for 24 h before treatment with or without 0.5 µM daunorubicin (DNR) for an additional 24 h. Cells were then analyzed by intracellular flow cytometry. Data were collected as mean fluorescence intensity (MFI) levels corrected for a “no primary antibody” sample of pro-apoptotic BCL2-family proteins (*n* = 5). P1 represents the population determined by viable DNR untreated cells, but also visible in DNR-treated cells, with minimal changes of protein expression. P2 represents the population generated by DNR-treatment. (**B**) MFI levels corrected for a “no primary antibody” sample of anti-apoptotic BCL2-family proteins (*n* = 5). Data were compared using the paired-sample *t*-test for data pairs. ** *p* < 0.001.

**Figure 5 ijms-22-07264-f005:**
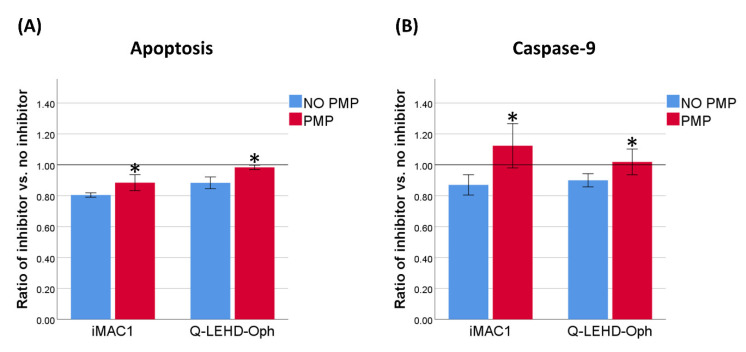
Effects of BAX and caspase-9 inhibitors on apoptosis and caspase-9 activation. (**A**) THP-1 cells were incubated with or without platelet microparticles (PMPs) for 23 h, and then with or without iMAC1 (BAX inhibitor), or Q-LEHD-Oph (caspase-9 inhibitor), for 1 h before treatment with 0.5 µM daunorubicin (DNR). Ratio of relative frequency of dead and apoptotic cells with or without inhibitor was calculated after 24 h *(n* = 3). (**B**) Ratio of relative frequency of caspase-9^mid/hi^ cells after 24 h of DNR-treatment with or without inhibitor (*n* = 3). Data were compared using the paired-sample *t*-test for data pairs. * *p* < 0.05.

**Figure 6 ijms-22-07264-f006:**
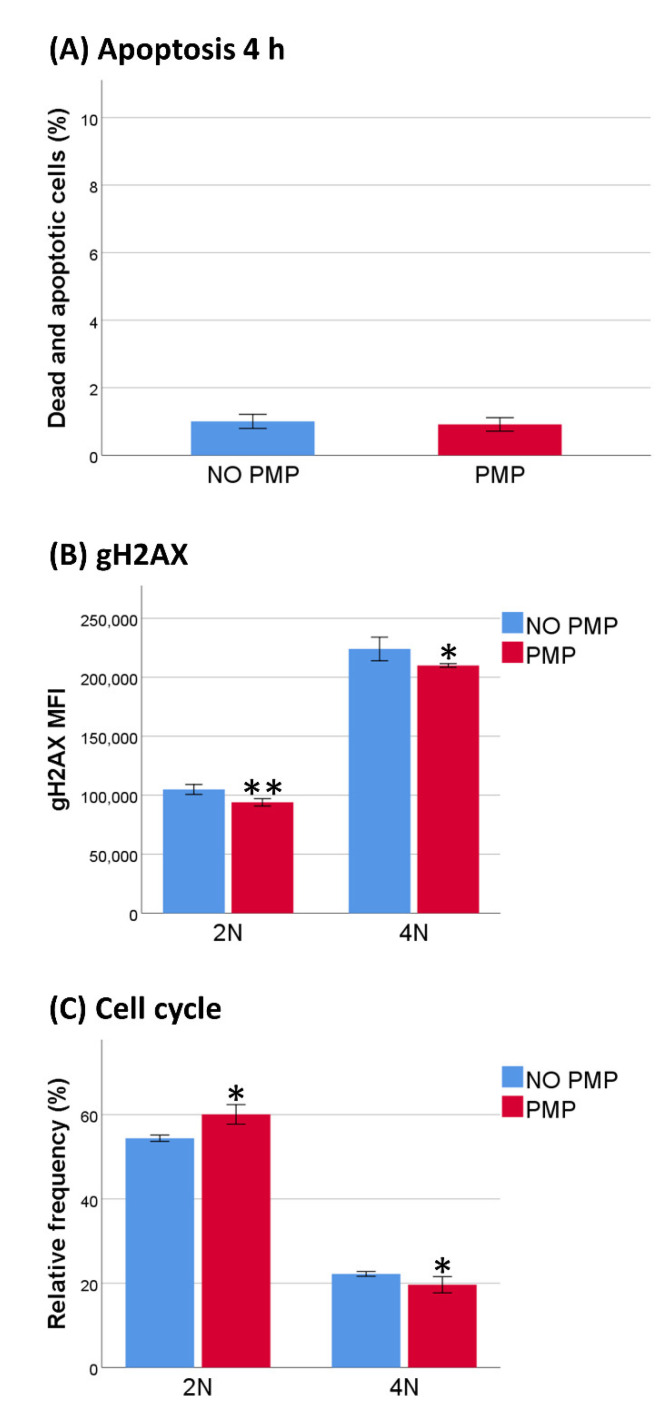
DNA damage after daunorubicin (DNR) treatment. (**A**) THP-1 cells were co-incubated with or without platelet microparticles (PMPs) for 24 h and treated with 0.5 µM DNR for 4 h before analysis with flow cytometry. Relative frequency of dead and apoptotic cells (*n* = 3). (**B**) Mean fluorescence intensity (MFI) of gH2AX corrected for both an unstained sample and background corrected MFI of representative experiment without DNR (*n* = 4). (**C**) Relative frequency of 2N (G1) and 4N (G2/M) cells (*n* = 4). Data were compared using the paired-sample *t*-test for data pairs. * *p* < 0.05.

## Data Availability

The data presented in this study are available on reasonable request from the corresponding author.

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
