# Peer review of "Platelet Microparticles Decrease Daunorubicin-Induced DNA Damage and Modulate Intrinsic Apoptosis in THP-1 Cells"

_ijms, 2021, doi:10.3390/ijms22147264_

Round 1
Reviewer 1 Report
This study using gating strategy for intracellular flow cytometry analysis identifies that the PMP from platelet enables the recipient THP-1 AML cell a decreased gH2AX (DSB) and PUMA abundance with the treatment of DNR. This is an interesting work. There are concerns on the manuscript and findings.
- 3 legend has been fused with section 3.4. This need to be improved by delicate edition.
- 2C needs to be presented in a manner to include no treatment and treatment settings.
- MFI and Mean Fluorescence Intensity are mixed uses in the Y-axis of figures. Please unify.
- The eminent decrease of DNR induced PUMA due to PMP needs to be confirmed by WB or immunofluorescence (IF) image.
- The mild decrease of gH2AX 4h after DNR treatment due to PMP needs to be confirmed by IF or more time courses.
- Discussion should encompass the requirement of more leukemia cell lines or patient’s derived tumor cells to yield firm conclusion. In addition to miR-125 families, molecules, cytokines or other non-coding RNAs in PMP, potent in apoptosis or drug sensitivity regulation should also be discussed.
Reviewer 2 Report
In the current study, the authors investigated the PMPs protection in AML cells against apoptosis after treatment with a selection of inducers, primarily associated with either the intrinsic or the extrinsic apoptotic pathway.
It is an interesting and well-conducted study. The introduction fully described the state of the art and the material and methods are clear. In the manuscript, the results are clearly described and convincing.
I suggest acceptabling for publication after minor point:
In Fig 1 the authors should review the p value (* p)
Round 2
Reviewer 1 Report
The paper is acceptable in its present form.
Reviewer 2 Report
Accept in present form